# EmbedFormer: Embedded Depth-Wise Convolution Layer for Token Mixing

**DOI:** 10.3390/s22249854

**Published:** 2022-12-15

**Authors:** Zeji Wang, Xiaowei He, Yi Li, Qinliang Chuai

**Affiliations:** College of Mathematics and Computer Science, Zhejiang Normal University, Jinhua 321004, China

**Keywords:** deep learning, computer vision, CNN, vision transformer

## Abstract

Visual Transformers (ViTs) have shown impressive performance due to their powerful coding ability to catch spatial and channel information. MetaFormer gives us a general architecture of transformers consisting of a token mixer and a channel mixer through which we can generally understand how transformers work. It is proved that the general architecture of the ViTs is more essential to the models’ performance than self-attention mechanism. Then, Depth-wise Convolution layer (DwConv) is widely accepted to replace local self-attention in transformers. In this work, a pure convolutional "transformer" is designed. We rethink the difference between the operation of self-attention and DwConv. It is found that the self-attention layer, with an embedding layer, unavoidably affects channel information, while DwConv only mixes the token information per channel. To address the differences between DwConv and self-attention, we implement DwConv with an embedding layer before as the token mixer to instantiate a MetaFormer block and a model named EmbedFormer is introduced. Meanwhile, SEBlock is applied in the channel mixer part to improve performance. On the ImageNet-1K classification task, EmbedFormer achieves top-1 accuracy of 81.7% without additional training images, surpassing the Swin transformer by +0.4% in similar complexity. In addition, EmbedFormer is evaluated in downstream tasks and the results are entirely above those of PoolFormer, ResNet and DeiT. Compared with PoolFormer-S24, another instance of MetaFormer, our EmbedFormer improves the score by +3.0% box AP/+2.3% mask AP on the COCO dataset and +1.3% mIoU on the ADE20K.

## 1. Introduction

Since AlexNet [1] showed extreme performance on the ImageNet image classification challenge [2], Convolutional Neural Networks (CNNs) have long been the most dominant method for modeling in computer vision because of their high efficiency in processing images. For example, when it comes to the backbones of vision tasks, such as image classification, object detection and tracing, ResNets [3] would always be the first thought because of its outstanding contribution to deep neural networks and excellent performance. There is no doubt that many novel model populations have been created to improve task performance by a large margin. At the same time, traditional CNNs are still not out of fashion, showing the strong ability of CNNs to learn sophisticated and complicated representations.

The transformer [4] was first designed for sequence modeling and transduction tasks in natural language processing and was introduced to computer vision, bringing in a crazy wave of Visual Transformers (ViTs) in recent years. Since then, transformer models have been successfully used for all kinds of vision tasks, such as image classification [5,6,7,8], object detection [9,10], semantic segmentation [11] and so on. The transformer has also been introduced to practical applications such as the disparity prediction algorithm [12].

Initially, researchers thought highly of self-attention, a novel mechanism for processing long-term semantic information in transformers. ViTs [5] treat pictures as 16×16 words and use a pure transformer to process them. It achieves good results while suffering from high computational costs. To deal with its substantial computational complexity, researchers have introduced many kinds of local self-attention (LSA) [6,7]. They no longer compute global attention but catch the neighboring relationships through ideas of small windows. It is worth mentioning that ViTs, the outputs of which seem like a columnar structure, were designed to deal with image classification tasks specifically. Since PVT [8], transformer-based backbones have started producing hierarchical representations that seem like a pyramid.

As time went by, it was proved that the general architecture of transformers is more essential to the models’ performances. Yu et al. [13] abstracted the transformer and its variants to a general architecture named MetaFormer, in which SA is merely regarded as one kind of token mixer. Then, many simple operators, such as Depth-wise Convolution (DwConv) [14], dynamic filter and even parameter-less pooling, are introduced for token mixing [13,15], showing that DwConv can match the performance of LSA [15]. Liu et al. [16] gradually modernized an original ResNet toward a transformer-like one named ConvNext, which achieves competitive performance with transformer-based backbones. ConvNext can be viewed as a simple visual transformer not using self-attention but DwConv to catch spatial information. The appearance of ConvNeXt is powerful proof that well designed CNNs are also competitive with visual transformers.

However, there are still several differences between transformer-based architectures and ConvNeXt. On the one hand, in terms of the block structure, visual transformer block has two shortcuts, while ConvNeXt block has only one. This means that visual transformer block processes an input through two separate steps, but ConvNeXt treats token mixing and channel mixing as one indispensable flow. On the other hand, DwConv has different operating logic for processing feature maps with a self-attention mechanism. The self-attention mechanism processes spatial and channel information, while DwConv only operates on the spatial dimension.

We consider building a pure convolutional architecture more like a transformer-based one in this work. To realize this idea, a convolutional token mixer which is efficient and more similar to the self-attention mechanism is designed and applied into a MetaFormer block to combine the advantages of both CNN and transformers. The proposed model is named EmbedFormer. The basal block of the proposal is sliced into two parts, the token mixing part and the channel mixing part, both with a shortcut beside them. Similar to ConvNeXt, DwConv is used as the token mixer. Differently, an embedding layer (point-wise convolution) is added to the head of the block to make the operation of DwConv more like self-attention. The token mixer is designed to avoid the computational complexity of self-attention and equalize the form with a transformer-based block. In the channel mixer, SEBlock [17] is introduced to improve the performance of the channel mixing part. The stage design of PoolFormmer-S24 is exploited, specifically stage depths of (4, 4, 12, 4) and stage channels of (64, 128, 320, 512). EmbedFormer is of similar computational complexity to ResNet-50 and surpasses its performance.

In this paper, firstly, the background and the motivation of EmbedFormer are shown in Section 1. Then, in Section 2, the related works, including CNN-based architectures, transformer-based architectures and attention modules, are introduced. In Section 3, the proposed model named EmbedFormer is displayed in detail. In Section 4, we evaluate the proposed EmbedFormer on several vision tasks, including ImageNet classification [2], object detection and segmentation on COCO [18] and semantic segmentation on ADE20K [19]. In Section 5, the ablation experiments are shown. Finally, in Section 6, we summarize the whole paper and put forward the future prospect.

## 2. Related Work

### 2.1. CNN-Based Architecture

CNN is one of the most significant networks in the deep learning field. It has been several decades since CNN was invented. However, when AlexNet [1] was born, CNNs became the soul of vision tasks because of their strong feature extraction ability. With the development of deep learning, CNNs became deeper and more effective. The one we are most familiar with is ResNet [3]. It has been the most widely used CNN in many tasks. Some variants of ResNet, such as ResNeXt [14] and RegNet [20], also achieve good results. Another well known series is EfficientNet [21] and its variants. In mobile applications, lightweight CNNs, such as MobileNet [22] and ShuffleNet [23], also take a place. CNNs usually serve as the backbone of a model in vision tasks.

Except for the advances of architecture, several works are aiming to improve individual convolution layers, such as DwConv [14]. DwConv is a particular case of grouped convolution [1] with group number equaling channel number. In other words, the filters in DwConv are not shared between every two channels. In addition, deformable convolution [24,25] and dilate convolution [26] are also notable improvements.

### 2.2. Transformer

In this section, we introduce the original transformer. Firstly, we introduce the self-attention mechanism and the variants in Section 2.2.1. Then, the architecture of the original transformer is shown in Section 2.2.2.

#### 2.2.1. Self-Attention

The transformer attention mechanism normally maps input sequences *X* and *Y* into three different sequential vectors (query *Q*, key *K* and value *V*) as follows:(1)$$Q=XW^{Q},K=YW^{K},V=YW^{V}$$
$W^{Q}$, $W^{K}$ and $W^{V}$ denote linear matrices. In a self-attention mechanism, where the two inputs are equivalent to each other, an embedding layer (often implemented with a point-wise convolution layer or a linear layer) is used to generate *Q*, *K* and *V*.

**Self-Attention** As the global information catcher, the self-attention mechanism is an integral component of the original transformer. The structure of self-attention is shown in Figure 1. The following equations show the operation of self-attention: (2)Q,K,V=Embedding(Input)
(3)$$Attn\left(Q,K,V\right)=Softmax\left(QK^{T}\right)V$$

The Attn() in Equation (Equation 3) denotes the output of the attention mechanism. The embedding layer in Equation (Equation 2) might be convolution or linear layers in implementation. In the self-attention mechanism, it is used to transfer the input sequence into query, key and value.

**Masked Self-Attention** A self-attention layer updates the input sequence by computing the relationships between every two components. It is through this mechanism that transformers have the capacity to aggregate the global information of the complete input. However, in the decoder, which is used to predict sequence components one by one, the following information of each component should be masked. Then, masked self-attention is employed and the following equations show the its operation: (4)Q,K,V=Embedding(Input)
(5)$$Attn\left(Q,K,V\right)=Softmax\left(QK^{T}⊙M\right)V$$

The mask *M* is an upper-triangular matrix and ⊙ denotes the Hadamard product. When predicting an entity in the sequence, the future entities’ attention score is set to zero in masked self-attention.

**Multi-Head Self-Attention** Self-attention can be improved by the idea of the multi-head mechanism. The input sequence is divided into several slices by the domain of channels and self-attention is computed during each slice. When *h* denotes the number of heads and $input^{\prime }$ denotes the set of ${\left\{input_{i}\right\}}_{i=1}^{h}$ in which every $input_{i}$ is a result of dividing the input sequence into *h* equal parts along the channel dimension, the following equations show the operation of multi-head self-attention: (6)$$Q^{\prime },K^{\prime },V^{\prime }=Embedding\left(input^{\prime }\right)$$
(7)$$head_{i}=Softmax\left(Q_{i}K_{i}^{T}\right)V_{i}$$
(8)$$MultiHeadAttn\left(Q^{\prime },K^{\prime },V^{\prime }\right)=Concat\left(head_{1},\ldots ,head_{h}\right)W^{o}$$

In the equations, $Q^{\prime }$ denotes the set of ${\left\{Q_{i}\right\}}_{i=1}^{h}$ (so does $K^{\prime }$, $V^{\prime }$). Concat is the operation of concatenation. $W^{o}$ is a linear projection matrix. It is said that every head aggregates different information from the entire input sequence.

#### 2.2.2. Original Transformer

The transformer [4] is a model firstly designed for Natural Language Processing (NLP) tasks and has been one of the most widely-used models in machine translation [27], speech question answering [28] and many other language tasks. The structure of the original transformer is shown in Figure 2, in which a feed-forward network is a two-layer Multi-Layer Perceptron (MLP). An original transformer has an encoder–decoder structure. An encoder consists of six encoder layers in which multi-head self-attention and forward feed network are employed with residual connections. Similar to the encoder, the decoder also has six decoder layers. Each decoder layer has three sub-layers, one masked multi-head self-attention and another two the same as the encoder layer. It is worth mentioning that the attention mechanism has no sense of positional information, so cosine positional encoding is implemented to add the positional vector into the input sequences.

### 2.3. Transformer-Based Architecture

Because of the success of transformers in NLP, researchers have attempted to use transformers in CV tasks, including image classification [5,6,7,8], object detection [9,10] and semantic segmentation [29].

ViT [5] firstly treats an image as 16×16 patches and introduces an original transformer model to process them. It attains state-of-art performance on image classification tasks with a large dataset for pre-training. Since the successful attempt of ViT, transformer-based backbone architectures have sprung up like mushrooms. Some of them take the original self-attention layer as ViT does, such as PVT [8], despite its unsatisfactory computational complexity. Many others use local self-attention mechanisms instead to speed up, like Swin transformer [6] and CSwin transformer [7]. Nowadays, transformer-based backbones often perform no worse than CNN-based ones. It is widely accepted that transformer-based backbones would perform better than CNN-based ones if pre-trained with larger datasets.

Yu et al. [13] proved that the brilliant achievement of visual transformers is owed to the well designed structure rather than the self-attention mechanism. They abstract the blocks of transformer-based backbones to a general one named MetaFormer, which consists of a token mixer and a channel mixer as shown in Figure 3a. It summarizes the workflow of a transformer block. When inputting an image (or feature map), it would first deal with the spacial information by every channel. Then, the channel information would be mixed by the channel mixing part.

### 2.4. Turning a CNN to a Transformer-like One

When the tricky structure of some local self-attention mechanisms was formed, such as shift-window self-attention introduced by the Swin transformer, the cost of implementation became expensive. In contrast, CNNs have been long known to be implementation-friendly. ConvNeXt [16] is a pure convolution neural network that benefits a lot from transformer-based backbones. By “modernizing” a ResNet-50 to a transformer-like one, it achieves better results than Swin transformer. Figure 3b shows the structure of a ConvNeXt block. As seen in this architecture, if we treat the 7×7 DwConv as a token mixer and the following layers as a channel mixer, a ConvNeXt can be regarded as a simple transformer with only one shortcut. The success of ConvNeXt proves the excellent power of convolution neural networks. When researching transformer-based architecture worldwide, the pure convolution architecture inspires us a lot.

### 2.5. Attention Module

The attention mechanism in computer vision is introduced to help a model rank the importance of information and naturally focuses on the more important sort. It can be employed for various visual models to improve the performance on almost all tasks, for example, the Single-Shot Object Detection (SSD) model on object detection [30]. Generally, in image tasks, the domains of their concern can be classified into three types: spatial, channel and mixed. Squeeze-and-Excitation Block (SEBlock) [17], whose structure is shown in Figure 4, is a standard attention module on the channel domain. It can improve the quality of representations produced by a network by explicitly modeling the inter-relationships between the channels, raising the results only by adding a small amount of computational cost. Following this work, many variants of channel attention modules come up, such as Selective Kernel Block [31], Coordinate Attention [32] and Style-based Recalibration Module [33].

## 3. EmbedFormer

In this section, the proposed model is introduced in detail. Firstly, the overall architecture is recommended in Section 3.1. Then, the structure of EmbedFormer block is shown in Section 3.2.

### 3.1. Overall Architecture

Figure 5 shows the general structure of EmbedFormer. Like Swin transformers and many other transformer-based networks, a patchify stem is introduced in EmbedFormer to process input images. This means that an input RGB image would be split into non-overlapping patches, which are treated as “tokens”. In the implementation, 4×4 convolution with stride 4 is used to separate the images into patches.

Four stages would process the tokens consisting of several EmbedFomer blocks. Between stages, downsample layers are utilized to decrease the number of tokens and expand the output dimension. 3×3 convolution with stride 2 is simply used for downsampling. As a result, a hierarchical representation would be produced and can be conveniently used in downstream tasks such as object detection and semantic segmentation.

### 3.2. EmbedFormer Block

Inspired by MetaFormer and ConvNeXt, the first designed block is shown in Figure 6a. It follows MetaFormer to be composed of two parts: the token-mixer and the channel-mixer parts. DwConv is used to play the role of token-mixer. Concerning the channel-mixer, it follows ConvNeXt, implementing only one layer norm and one GELU as the activation function.

Then, we reflect on the pattern of self-attention and DwConv. As is widely accepted, DwConv performs as well as local self-attention and is thought to have the equivalent operation of the latter. However, it is found that different from DwConv, the self-attention mechanism has an embedding layer before computing the attention. It is unavoidable to cause an effect on the channel information. To compensate for this, we add an embedding layer before the DwConv layer. The following equations show the operation of our token mixer: (9)X=Embedding(Input)
(10)Output=DwConv(X)

To help stabilize training and enhance the nonlinearity of the model, we add one layer-norm and one GELU in the token mixer. Until now, the structure of the block is shown in Figure 6b.

In addition, channel-wise attention like SEBlock still contributes to helping the channel-mixer catch information. The final structure of the block is shown in Figure 6c. It is worth mentioning that removing the ReLU function between the two point-wise convolution layer would make the model perform better. The modified SEBlock is shown in Figure 7.

## 4. Experiments and Results

In this section, the experiments and results are shown in tables. An EmbedFormer of stage depths (4, 4, 12, 4) and channels (64, 128, 320, 512) is constructed to be of similar complexity to ResNet-50. We firstly train and value our model on the ImageNet-1K data set [2]. Then, the models are fine-tuned with pre-trained EmbedFormer backbone on COCO [18] and ADE20K [19] to test its applicability for downstream tasks, specifically object detection and semantic segmentation.

### 4.1. Image Classification on ImageNet-1K

ImageNet-1K, one of the most widely used datasets in visual classification, contains about 1.3 M training images and 50 K validation images, covering 1 K classes. The proposed model is trained for 300 epochs using AdamW optimizer [34] with weight decay 0.05, batch size 128 and peak learning rate 5 × 10${}^{-4}$. The number of linear warmup epochs is 20 with a cosine learning rate schedule. Meanwhile, typical schemes, including Mixup [35], Cutmix [36], RandAugment [37] and Random Erasing [38] are adopted for data augmentation as in [16]. The results are shown in Table 1.

As shown in the table, the proposed EmbedFormer achieves 81.7 top-1 accuracy, outperforming RSB-ResNet-50 [39], RegNetY-4G [20], DeiT-S [40] and some other models of similar complexity. As another instance of MetaFormer, EmbedFormer obtains a higher score (+1.4%) than PoolFormer-S24 on the ImageNet classification task. Compared with the Swin transformer, EmbedFormer performs 0.4% better on top-1 accuracy with a smaller model scale and less computational cost. However compared with ConvNeXt-T, the proposed EmbedFormer achieves −0.4% top-1 accuracy owing to a 6 million gap in the number of parameters.

We visualize the results of ResNet-50, DeiT-S, Swin-T and the proposed EmbedFormer by Grad-CAM [41]. The figures are shown in Figure 8. As is shown, EmbedFormer can accurately find the critical feature of an object. Besides the outstanding results on classification task, EmbedFormer also offers excellent performance when applied to the downstream vision tasks as a backbone.

### 4.2. Object Detection on COCO

Containing 118 K training and 5 K validation images, COCO is one of the most commonly used data sets in the object detection field. With EmbedFormer backbone, we fine-tune Mask R-CNN [42] and Cascade Mask R-CNN [43] on COCO. Mask R-CNN is trained using 1× schedule with single-scale training and AdamW optimizer as with PoolFormer [13]. Cascade Mask R-CNN is trained primarily following the Swin transformer [6]. In other words, multi-scale training, AdamW optimizer and 3× schedule are adopted. We train and value our model on mmdetection [44]. The results are shown in Table 2.

We first compare EmbedFormer with CNN-based ResNet-50. Both as the backbone in Mask R-CNN and Cascade R-CNN, EmbedFormer performs much better than ResNet-50 (+5% and +2.3 box AP). Both as instances of MetaFormer, the proposed EmbedFormer brings 2.9% box AP and 2.3% mask AP gains over PoolFormer with slightly larger model size and latency. Compared with mainstream transformer-based architecture DeiT-S, EmbedFormer achieves +0.6% box AP and +0.9% mask AP improvement at over twice the efficiency. EmbedFormer can also surpass Swin-T by +0.3% box AP when trained with 1× schedule.

### 4.3. Semantic Segmentation on ADE20K

We also try the semantic segmentation task on the widely used dataset, ADE20K, which covers a broad range of 150 semantic categories, with 20 K images for training and 2 K for validation. Semantic FPN [45] and UPerNet [46] are used as segmentors and mostly follow the settings of Poolformer [13] and Swin transformer [6], respectively. We train and evaluate the proposed model on mmsegmentation [47]. The results are shown in Table 3.

These results show that EmbedFormer is +4.9% (+2.5%) mIoU higher than ResNet-50 as the backbone in Semantic FPN (UPerNet) of similar model size. It is also +1.8% mIoU higher than PVT-Small and +1.3% mIoU higher than PoolFormer. Compared with DeiT-S, EmbedFormer is +1.3% mIoU higher with almost twice the inference speed. The proposed EmbedFormer achieves the same mIoU with Swin-T (41.6%) when trained with 80K iterations and surpasses it by +0.8% when trained with 160K iterations in UPerNet.

## 5. Ablation Experiments

To find a well designed architecture, we designed a set of blocks starting from a MetaFormer block with a DwConvn layer as the token mixer. For fast experiments, the designed models are trained with stage depths of (4, 4, 12, 4) and stage channels of (64, 128, 320, 512) on ImageNet dataset for only 30 epochs. Training on one piece of RTX3090, the batch size is set to 128 and the learning rate is set to 5 × 10${}^{-4}$. Cosine training schedule and AdamW optimization are implemented. The results are shown in Table 4.

Firstly, 7×7 DwConv is used as the token mixer, as in the structure shown in Figure 6a, achieving 74.98 top-1 accuracy. After the DwConv layer, we add a GELU function, seeing a temporary reduction in the accuracy. We find a fantastic enhancement when introducing the embedding layer (with a layer norm behind) before the DwConv. After that, we try adding SEBlock into the block. It performs better when placed in the second shortcut, before the channel mixer, as shown in Figure 6c. It is interesting to note that if the GELU function is removed behind the DwConv, the accuracy would drop dramatically.

## 6. Conclusions

In the past few years, transformer-based backbones, especially Swin transformer, have shown strong power concerning vision tasks, gradually attracting more eyes than CNNs. However, CNNs still hold an irreplaceable position in practical applications because of their convenience of deployment. Inspired by MetaFormer and ConvNeXt, we introduce EmbedFormer in this paper to explore the possibility of convolution layers taking the place of the spatial mixer. As a result, EmbedFormer is evaluated on several shared vision tasks such as ImageNet classification [2], object detection/segmentation on COCO [18] and semantic segmentation on ADE20K [19]. In ImageNet classification, EmbedFormer achieves 81.7 top-1 accuracy without extra training images, surpassing Swin transformer by 0.4%. Box AP/mask AP 48.6/42.3 on COCO and 44.3 of mIoU on ADE20K also significantly adapt downstream vision tasks. EmbedFormer achieves better performance on almost all vision tasks compared to Swin transformer. The proposed EmbedFormer proves that it is possible to design a pure convolutional “transformer” on vision tasks. There might be space for improvement for EmbedFormer. We believe more efficient convolutional “transformers” will be proposed in the near future.

## Figures and Tables

**Figure 1 sensors-22-09854-f001:**
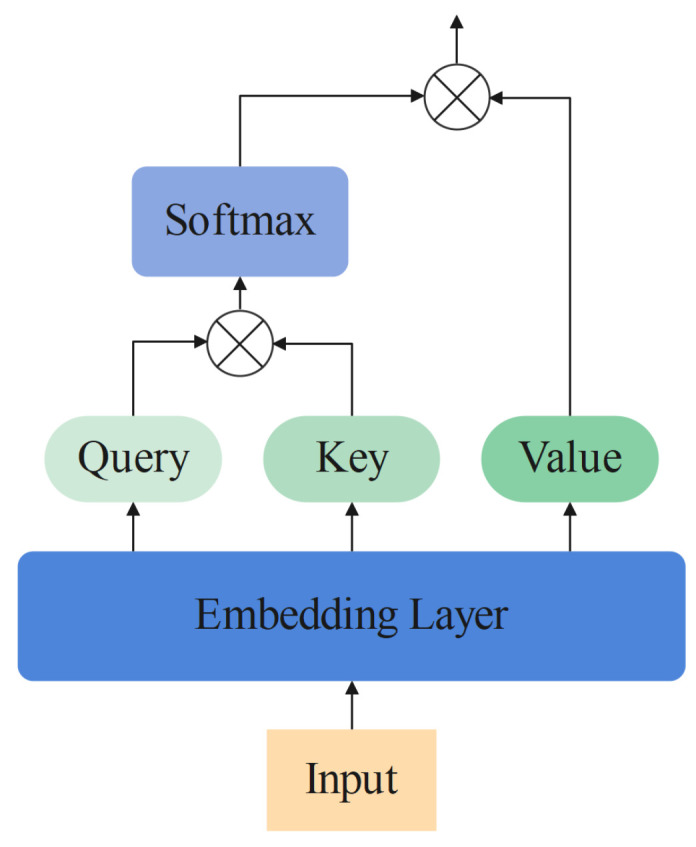
The computing operation of self-attention.

**Figure 2 sensors-22-09854-f002:**
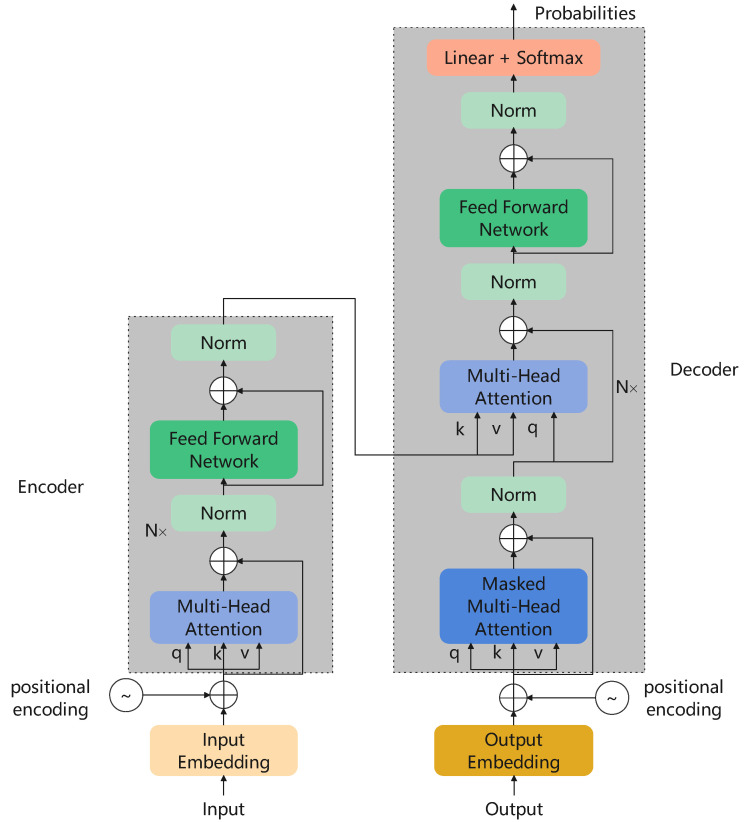
The structure of the original transformer.

**Figure 3 sensors-22-09854-f003:**
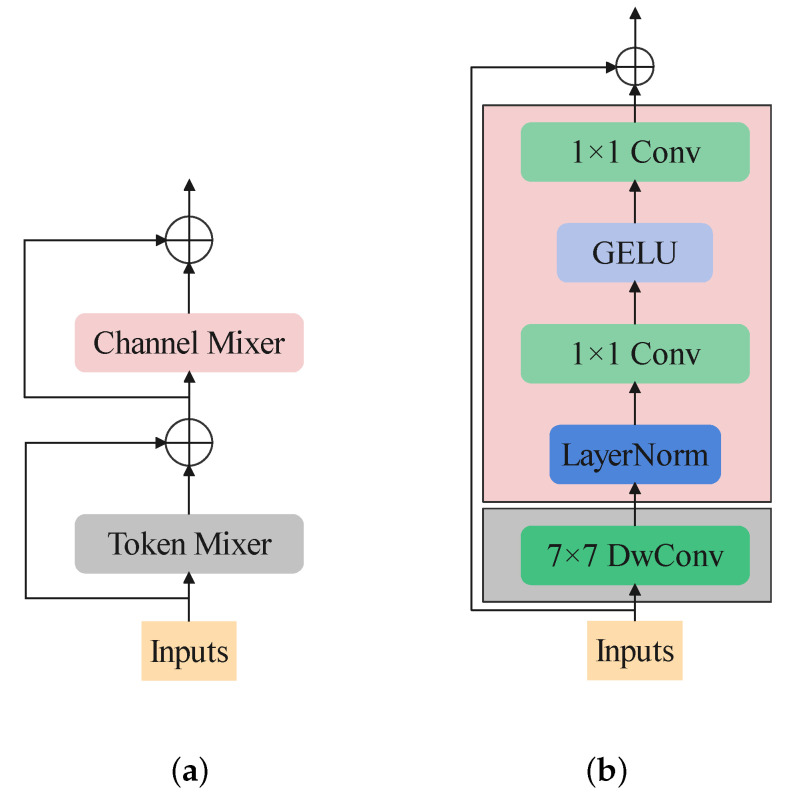
(**a**) The structure of MetaFormer block. (**b**) The structure of ConvNeXt block.

**Figure 4 sensors-22-09854-f004:**
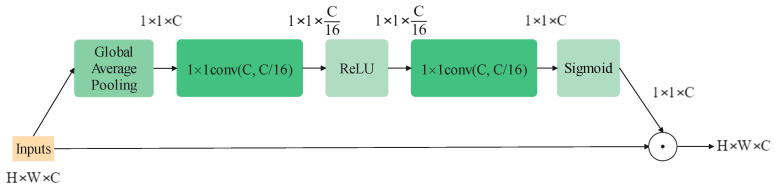
The structure of an SEBlock.

**Figure 5 sensors-22-09854-f005:**
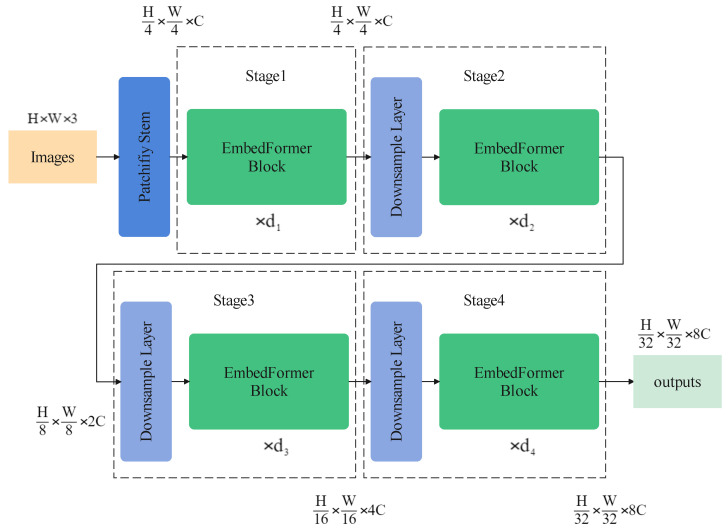
The overall architecture of an EmbedFormer.

**Figure 6 sensors-22-09854-f006:**
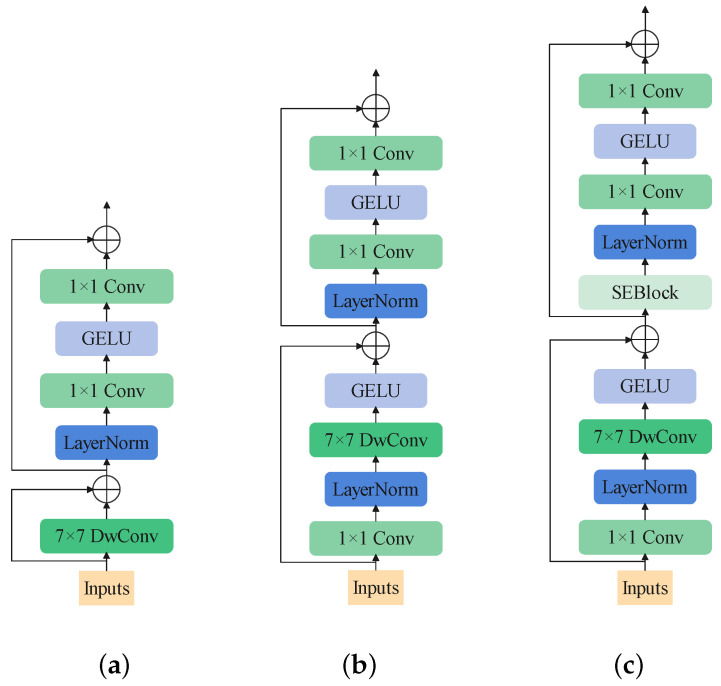
The order of designing the EmbedFormer block. (**a**) The structure of the first designed block. (**b**) The structure of the second designed block. (**c**) The structure of EmbedFormer block.

**Figure 7 sensors-22-09854-f007:**
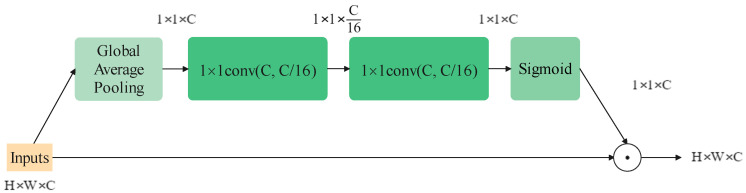
The modified SEBlock structure used in the proposed model.

**Figure 8 sensors-22-09854-f008:**
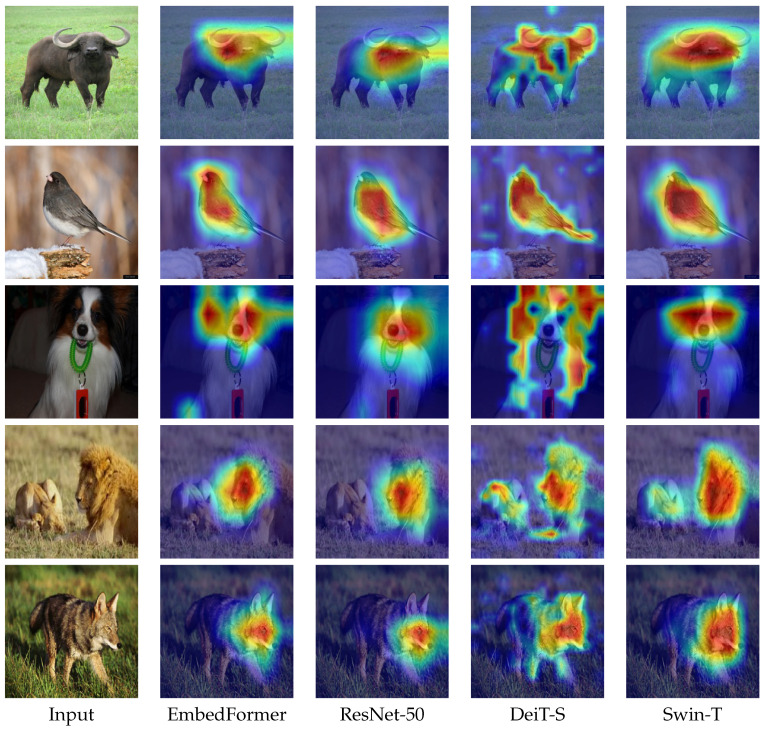
The Grad-CAM activation maps of the models trained on ImageNet-1K.

**Table 1 sensors-22-09854-t001:** The results of EmbedFormer and different mainstream backbones of similar complexity on ImageNet-1K.

Model	Param.	MACs	Top-1 Acc.
RSB-ResNet-50 [39]	26 M	4.1 G	79.8
RegNetY-4G [20]	21M	4.0 G	80.0
ViT-B [5]	86 M	55.4 G	77.9
DeiT-S [40]	22 M	4.6 G	79.8
PVT-Small [8]	25 M	3.8 G	79.8
PoolFormer-S24 [13]	21 M	3.6 G	80.3
Swin-T [6]	29 M	4.5 G	81.3
ConvNeXt-T [16]	29 M	4.5 G	82.1
EmbedFormer	23 M	3.8 G	81.7

**Table 2 sensors-22-09854-t002:** The object detection and segmentation results on COCO using Mask R-CNN and Cascade Mask R-CNN. The results of FPS are tested on one piece of RTX-3090.

**Mask R-CNN (**1× **Schedule)**								
backbone	${\mathrm{AP}}^{box}$	${\mathrm{AP}}_{50}^{box}$	${\mathrm{AP}}_{75}^{box}$	${\mathrm{AP}}^{mask}$	${\mathrm{AP}}_{50}^{mask}$	${\mathrm{AP}}_{75}^{mask}$	#param	FPS
ResNet-50 [3]	38.0	58.6	41.4	34.4	55.1	36.7	44M	26.3
PoolFormer-S24 [13]	40.1	62.2	43.4	37.0	59.1	39.6	41M	28.0
Swin-T [6]	42.7	64.9	46.7	39.4	61.8	42.2	48M	22.9
EmbedFormer	43.0	64.3	47.0	39.3	61.6	42.0	43M	25.1
**Cascade Mask R-CNN (** 3× **Schedule)**								
backbone	${\mathrm{AP}}^{box}$	${\mathrm{AP}}_{50}^{box}$	${\mathrm{AP}}_{75}^{box}$	${\mathrm{AP}}^{mask}$	${\mathrm{AP}}_{50}^{mask}$	${\mathrm{AP}}_{75}^{mask}$	#param	FPS
ResNet-50 [3]	46.3	64.3	50.5	40.1	61.7	43.4	82M	18.7
DeiT-S [40]	48.0	67.2	51.7	41.4	64.2	44.3	80M	8.0
Swin-T [6]	50.5	69.3	54.9	43.7	66.6	47.1	86M	11.7
EmbedFormer	48.6	66.9	52.7	42.3	64.6	45.8	81M	17.5

**Table 3 sensors-22-09854-t003:** The semantic segmentation results on ADE20K using Semantic FPN and UPerNet. The results of FPS are tested on one piece of RTX-3090.

**Semantic FPN (80 K Iterations)**			
backbone	val mIoU	#param	FPS
ResNet-50 [3]	36.7	29 M	63.7
PVT-Small [8]	39.8	28 M	31.0
PoolFormer-S24 [13]	40.3	23 M	74.3
Swin-T [6]	41.6	32 M	40.8
EmbedFormer	41.6	27 M	49.9
**UPerNet (160 K Iterations)**			
backbone	val mIoU	#param	FPS
ResNet-50 [3]	42.8	67 M	32.6
DeiT-S [40]	44.0	52 M	18.0
Swin-T [6]	44.5	60 M	30.4
EmbedFormer	45.3	60 M	30.4

**Table 4 sensors-22-09854-t004:** The performance results of the designed models on ImageNet. The symbol ✓ indicates that this module is used in this design.

Embedding Layer	GELU	SEBlock	Top-1 Acc.	Top-5 Acc.
✓	✓	in the second shortcut	**76.13**	**92.91**
✓	✓	between shortcuts	75.68	92.79
✓	✓		76.04	92.90
✓		in the second shortcut	75.46	92.74
	✓		74.72	92.18
			74.98	92.31

## Data Availability

Some or all data, models or code generated or used during the study are available from the corresponding author by request.

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
