# Peer review of "EmbedFormer: Embedded Depth-Wise Convolution Layer for Token Mixing"

_sensors, 2022, doi:10.3390/s22249854_

Round 1
Reviewer 1 Report
This paper proposes a variation of the Transformer architecture in the line of Metaformers and ConvNext. It certainly has some positive points, but also some important drawbacks that require a revision process.
Regarding the positive aspects:
- The paper addresses a very interesting and trending topic.
- The experiment setup is quite exhaustive.
Regarding the negative aspects to be revised, the most relevant are:
- English should be (largely) improved, sometimes it is not possible to understand the narrative well.
- The contribution is just a slight change in the architecture of existing works, such as Transformers, Metaformers, or ConvNext, whose performance does not seem clearly better (see next point).
- The results do not seem to suggest that the proposal is better than others. From tables 2 and 3, Swin-T is in general better than the proposed EmbedFormer and also faster. Related to this, the result discussion only compares the proposed EmbedFormer with other alternatives (such as ResNet) that seem to be somewhat worse, but no comparison or serious analysis is made with the best one, Swin-T.
- ConvNext is used as inspiration for this work, but it does not appear in the comparison results. I really think that it is very convenient to include it.
- I miss some kind of activation function at the end of the last stage of EmbedFormer, but according to Fig 5 and 6 it does not exist.
- It is claimed that including an embedding layer before the depth-wise layer is convenient to achieve a comparable performance to the self-attention mechanism. However, this is already performed in the initial 4x4 convolution that patchify the image input. That convolution combines (process) all the pixels in every block, likewise the embedding layer (via a dense layer) restricted to the elements of every block (or patch or token).
- The description of Masked Self-Attention should be improved.
Other minor aspects for revision are:
- Include a little of context in the section "2.2 Transformer", i.e. introduce at least the other subsections.
- The positional encoding of Transformer is not mentioned / explained, However, it appears in Fig. 2.
- Errata: per-trained --> pre-trained (line 144)
- In table 4, it is not very intuitive to mark with an X that a certain module is missing.
Reviewer 2 Report
In this paper, the author proposes a new visual transformer-like structure. The main idea is easy to understand.
1. The structure is more similar with ConvNeXt than MetaFormer. Therefore, the name will be more accurate, like EmbedConvNeXt.
2. This is not a weakness. I hope the author could add experiments or discussion on general image retrieval datasets, such as such as building retrieval [i] and drone-based geo-localization [j], which also focuses on shape. I also look forward the author can add more of your ideas of person reid to future works. I think the paper is good to be accepted, and more your ideas will make it better. Thanks.
[i] Ge Y, Wang H, Zhu F, et al. Self-supervising fine-grained region similarities for large-scale image localization[C]//European Conference on Computer Vision. Springer, Cham, 2020: 369-386.
[j]Zheng Z, Wei Y, Yang Y. University-1652: A Multi-view Multi-source Benchmark for Drone-based Geo-localization[C]//Proceedings of the 28th ACM international conference on Multimedia. 2020: 1395-1403.
3. Besides, it would be great if the author could add doi for every transaction papers by adding this line
Reviewer 3 Report
This manuscript proposes EmbedFormer to explore the possibility of using DwConv with an embedding layer taking the place of special mixer and tested on vision tasks with some better performance. This study is important for addressing the design and architecture of Transformers work. The core idea seems interesting, but the paper should be improved in some regards:
What is the main motivation of this study? What will the gap be addressed here? There is various architecture of Transformers that have been studied and used by various researchers. What is the framework proposed here by EmbedFormer? More explanation about the proposed model should be given. Please explain a few more works that are related to improving the Transformers model. What is the significant difference between this manuscript with other works? Clearly state which performance metrics is used.
Please avoid usage of first-person pronouns such as “I,” “We,” “ours,” “us” as much as possible to maintain the tertiary nature of this publication and maintain a neutral voice in the article. For instance, 'We' can be rephrased as 'the authors', ‘our EmbedFormer’ can be rephrased as ‘proposed EmbedFormer’, etc.
Title: Could use full name for DwConv.
Lines 69-76: This last paragraph should improve. Should explain the flow of the organization of the sections instead of giving the result of the evaluations.
Line 98: Should define Q, K, V and Attn.
Line 246: Caption Table 2. Please check, the other models are not only [13] and [6]. Could bold the best performance result. Prefer to remove the ‘ours’ in EmbedFormer.
Line 251: Please improve this sentence.
Line 273: Table 4. What is the meaning of ‘cross’ X? Should include ‘tick’ for added one?
Section 6: Prefer to have section Conclusion instead of Discussion and repeating the results here again. Should explain more in details if would like to include future work. Should highlight the conclusion/significant contribution and impact of this study.
Some captions for figures and tables should be rechecked and improved. Preferably to avoid lengthy caption and explanation. Also, should avoid use ‘we’.
References: Why bold for year in some references?
